# Outside any therapeutic trial prescription of hydroxychloroquine for hospitalized patients with covid-19 during the first wave of the pandemic: A national inquiry of prescription patterns among French hospitalists

**Antoine Bosquet**[1]*, **Comlan Affo**[1], **Ludovic Plaisance**[2], **Géraldine Poenou**[2], **Emmanuel Mortier**[3], **Isabelle Mahé**[2,4]

**1** Assistance-Publique–Hôpitaux de Paris (AP-HP), DMU ESPRIT, Service de Médecine Interne, Hôpital Louis-Mourier, Université de Paris, Colombes, France, **2** Université de Paris, Assistance-Publique–Hôpitaux de Paris (AP-HP), DMU ESPRIT, Service de Médecine Interne, Hôpital Louis-Mourier, Colombes, France, **3** Assistance-Publique–Hôpitaux de Paris (AP-HP), DMU ESPRIT, Policlinique, Hôpital Louis-Mourier, Colombes, France, **4** Inserm UMR_S1140, Innovative Therapies in Haemostasis Paris, Paris, France

* antoine.bosquet@aphp.fr

## Abstract

### Introduction

During the first wave of the coronavirus-disease 2019 (covid-19) pandemic in early 2020, hydroxychloroquine (HCQ) was widely prescribed in light of in vitro activity against severe acute respiratory syndrome–coronavirus-2 (SARS-CoV-2). Our objective was to evaluate in early 2020 the rate of French hospitalists declaring having prescribed HCQ to treat covid-19 patients outside any therapeutic trial, compare the reasons and the determinants for having prescribed HCQ or not.

### Material and methods

A national inquiry submitted by email from May 7 to 25, 2020, to a sample of French hospitalists: doctors managing patients hospitalized for covid-19 in a French department of internal medicine or infectious diseases and identified in the directories of French hospitals or as a member of the French Infectious Diseases Society (SPILF). Primary outcome was the percentage of hospitalists declaring having prescribed HCQ to covid-19 patients. Secondary outcomes were reasons and determinants of HCQ prescription.

### Results

Among 400 (22.8%) responding hospitalists, 45.3% (95% CI, 40.4 to 50.1%) declared having prescribed HCQ to covid-19 patients. Two main profiles were discerned: HCQ prescribers who did not raise its efficacy as a motive, and non-prescribers who based their decision on evidence-based medicine. Multivariate analysis retained the following prescription determinants (adjusted odds ratio; 95% confidence interval): a departmental procedure for HCQ

**Data Availability Statement:** All relevant data are within the paper and its Supporting Information files.

**Funding:** The author(s) received no specific funding for this work.

**Competing interests:** Dr. Mahé reports grants from BMS Pfizer, grants from LEO Pharma, personal fees and non-financial support from BMS Pfizer, personal fees and non-financial support from Leo Pharma, personal fees and non-financial support from Bayer, outside the submitted work. This does not alter our adherence to PLOS ONE policies on sharing data and materials.

prescription (8.25; 4.79 to 14.20), having prescribed other treatments outside a therapeutic trial (3.21; 1.81 to 5.71), prior HCQ prescription (2.75; 1.5 to 5.03) and HCQ prescribed within the framework of a therapeutic trial (0.56; 0.33 to 0.95).

## Conclusion

Almost half of the hospitalists prescribed HCQ. The physician's personality (questioning or not evidence-based–medicine principles in the context of the pandemic) and departmental therapeutic procedures were the main factors influencing HCQ prescription. Establishment of "therapeutic" procedures represents a potential means to improve the quality of therapeutic decision-making during a pandemic.

## Introduction

By March 2021, the coronavirus-disease 2019 (covid-19) pandemic had affected more than 120 million persons worldwide and had led to more than 2.5 million deaths [1]. Despite the rapid initiation of numerous therapeutic trials [2], no antiviral treatment had proven efficacy in 2020 [3]. Chloroquine and hydroxychloroquine (HCQ) were the first drugs proposed to treat covid-19 in light of their in vitro activity against severe acute respiratory syndrome–coronavirus-2 (SARS-CoV-2) [4] that causes covid-19. HCQ has been used for decades to treat malaria and autoimmune diseases with a good safety profile and is inexpensive. At the start of 2020, the results of some studies suggested that HCQ might be effective against SARS-CoV-2 [5,6]. In the emergency context and without proof of its efficacy, HCQ was recommended in the national policies of many countries, notably emerging nations [7]. HCQ use was the object of numerous debates among caregivers and the public at large [8,9]. Results of physician surveys [10–12], analysis of HCQ prescriptions filled in city pharmacies [13–15] and observational in-hospital studies (S1 Table) showed that HCQ was widely prescribed worldwide during the first wave of the covid-19 pandemic in early 2020. However, no specific data on doctors' reasons and determinants for prescribing HCQ in-hospital were available.

On 25 March 2020, French government authorized HCQ use only for hospitalized patients, after informed consent had been obtained and based on a collegial decision [16]. Because that authorization was in no way a recommendation, it remains to assess the attitudes and opinions of hospitalists.

The primary objective was to determine the percentage of internal medicine or infectious disease hospitalists declaring having started HCQ outside a therapeutic trial to manage covid-19 patients. Secondary aims were to analyze the reasons the hospitalists gave for having prescribed HCQ or not, and what determined that decision.

## Materials and methods

### Design of the study

**Questionnaire.** We built a questionnaire that was drafted, stored and available on Google Form© (S1-S3 Appendices). The first versions of the questionnaire were tested on departmental hospitalists (IM, LP, LA, JC, SD) to evaluate comprehension of the items and the time needed to complete the inquiry. The questionnaire consists of 68 questions: 25 were asked of all participants, 43 only to some according to their previous answers, with the total number

varying from 37 to 62. A link to the electronic questionnaire was sent by e-mail starting May 1, 2020, with reminders sent at 1-week intervals, and closure May 25, 2020.

**Participating hospitalists.** The population targeted was defined as hospitalists managing covid-19 patients and practicing in France in a department of internal medicine or infectious disease, and entered in the directories of French hospitals (n = 1387) [17,18] or members of the French Infectious Diseases Society (SPILF) (n = 572). After the exclusion of 80 duplicates, the questionnaire was sent to 1879 hospitalists. Each participant provided written consent prior to gaining access to the questionnaire. Participants were not paid.

## Declaration to the French Computer Watchdog Commission (CNIL)

The agreement of conformity of the study was obtained from the CNIL on April 20, 2020 (no. 2217633 v 0).

## Statistical methods

The analysis considered two groups of hospitalists: one was composed of those who reported having started HCQ for a patient outside a clinical trial at least once (henceforth called prescribers); the other group (non-prescribers) included physicians who declared not having initiated HCQ, except within the framework of a therapeutic trial.

Continuous variables are reported as mean ± standard deviation (SD). Categorical variables are reported as number (percentages; which were calculated excluding missing data). Every relevant proportion is accompanied by its 2-sided 95% confidence interval (Wilson method). Missing data were not handled. A logistic-regression model assessed HCQ-prescription determinants. First, univariate analyses (p<0.2) selected potential explanatory variables that were then entered into the multivariate model (stepwise method with entry/stay significance levels of 0.2/0.05). The results are expressed as adjusted odd ratios (aOR) with their 95% confidence intervals. Due to the heterogeneity of the numbers of physicians per region (6 regions had <5% of the sample), to analyze the variable region in the logistic-regression model, the 13 metropolitan regions (exclusion of the three physicians from overseas departments) were grouped into five geographical areas: Île-de-France, Northeast (Bourgogne-Franche-Comté, Grand Est, Hauts-de-France), Northwest (Normandy, Brittany, Centre-Val de Loire, Pays de la Loire), Southwest (Nouvelle-Aquitaine, Occitanie) and Southeast (Auvergne-Rhône-Alpes, Provence-Alpes-Côte d'Azur, Corsica). Pearson's correlation coefficient and its 95% confidence interval were used to assess the relationship between the HCQ-prescription rate and cumulative in-hospital–mortality rate per region. A p-value <0.05 was considered significant, unless specified otherwise. All statistical analyses were performed with SAS release 9.4 (SAS Institute Inc, Cary, NC) statistical software package.

## Results

Among the 1879 to whom the questionnaire was sent, 127 were not received (distribution error message received after sending), 400 hospitalists completed the entire questionnaire, for a response rate of 22.8% (400/1752).

## Responders' characteristics

The demographic characteristics of the responders who completed the questionnaire are reported in Tables 1 and S2.

**Table 1. Participating hospitalists' characteristics.**

| | HCQ prescription for covid-19 patients | | |
|---|---|---|---|
| Characteristic | Total, n (%) | Yes, n (%) | No, n (%) |
| Sex | | | |
| n | 400 | 181 | 219 |
| Male | 208 (52) | 87 (48.1) | 121 (55.3) |
| Female | 192 (48) | 94 (51.9) | 98 (44.7) |
| Years in practice, n | | | |
| n | 400 | 181 | 219 |
| 0–4 | 66 (16.5) | 30 (16.6) | 36 (16.4) |
| 5–9 | 76 (19) | 32 (17.7) | 44 (20.1) |
| 10–19 | 91 (22.8) | 43 (23.8) | 48 (21.9) |
| 20–29 | 99 (24.8) | 51 (28.2) | 48 (21.9) |
| ≥30 | 68 (17) | 25 (13.8) | 43 (19.6) |
| Hospital type | | | |
| n | 400 | 181 | 219 |
| Private | 22 (5.5) | 12 (6.6) | 10 (4.6) |
| Teaching public | 210 (51.5) | 89 (49.2) | 121 (55.3) |
| Non-teaching public | 168 (42) | 80 (44.2) | 88 (40.2) |
| Specialty | | | |
| n | 400 | 181 | 219 |
| Other specialties | 91 (22.8) | 33 (18.2) | 58 (26.5) |
| Infectious diseases (ID) | 144 (36) | 58 (32.0) | 86 (39.3) |
| Internal medicine (IM) | 137 (34.3) | 73 (40.3) | 64 (29.2) |
| ID & IM | 28 (7) | 17 (9.4) | 11 (5.0) |
| Hospital region | | | |
| n | 400 | 181 | 219 |
| Auvergne Rhône-Alpes | 37 (9.3) | 21 (11.6) | 16 (7.3) |
| Bourgogne-Franche-Comté | 24 (6.0) | 15 (8.3) | 9 (4.1) |
| Brittany | 10 (2.5) | 0 (0.0) | 10 (4.6) |
| Centre-Val de Loire | 16 (4.0) | 6 (3.3) | 10 (4.6) |
| Grand Est | 58 (14.5) | 19 (10.5) | 39 (17.8) |
| Hauts-de-France | 34 8.5) | 7 (3.9) | 27 (12.3) |
| Île-de-France | 109 (27.3) | 61 (33.7) | 48 (21.9) |
| Normandy | 20 (5.0) | 14 (7.7) | 6 (2.7) |
| Nouvelle-Aquitaine | 24 (6.0) | 10 (5.5) | 14 (6.4) |
| Occitanie | 18 (4.5) | 7 (3.9) | 11 (5.0) |
| Pays de la Loire | 18 (4.5) | 3 (1.7) | 15 (6.8) |
| Provence-Alpes-Côte d'Azur & Corsica | 29 (7.3) | 17 (9.4) | 12 (5.5) |
| Overseas departments | 3 (0.8) | 1 (0.6) | 2 (0.9) |
| Previous HCQ prescription | | | |
| n | 398 | 181 | 217 |
| No | 89 (22.4) | 28 (15.5) | 61 (28.1) |
| Yes | 309 (77.6) | 153 (84.5) | 156 (71.9) |

HCQ was started to treat covid-19 by 45.3% of the hospitalists (95% CI, 40.4–50.1%); 93.6% (205/219) of the remaining responders never prescribed HCQ to their covid-19 patients, but 6.4% (14/219) sometimes continued HCQ prescribed by colleagues.

## Physicians reasons for prescribing HCQ or not

Hospitalists' most frequently chosen reasons for prescribing HCQ (Table 2) were the only therapeutic option available (no alternative; 56.9%), application of a collegial decision (50.8%), the favorable HCQ safety profile (49.7%), the potential severity of covid-19 (48.6%). About a third of the hospitalists recognized the uncertainty of HCQ efficacy, while only 13.8% declared having prescribed HCQ because "HCQ seemed effective". For 10% (n = 19, 95% CI, 6.8 to 15.8%) of the responders, the prescription continued a third-party decision, the only reason for prescription was either "I applied a collegial decision" or "It was requested by the patients and/or his/her entourage" (n = 4).

Overall, 84.5% (95% CI, 79.1 to 88.7%) of non-prescribers justified their choice by at least one of the following reasons (Table 2): "No indication according to the available medicine/science-based" (74.4%), "Consider it unethical to prescribe a drug that is not validated outside therapeutic trials" (35.6%), "opposed to off-label prescription" (4.6%) "No official recommendations supporting HCQ prescription" (50.7%). The remaining non-prescribers (15.5%) had practical reasons: no collegial discussion was organized in their department or patients did not meet the criteria for prescription. In addition, ~36% of non-prescribers feared potential adverse events of HCQ or covid-19 worsening under HCQ (5.5%).

## Determinants of HCQ prescription

Univariate analyses identified factors associated with HCQ prescription (Tables 3 and S3). The most important was an established departmental procedure that increased HCQ prescriptions (8.36 [5.12 to 13.65]). That protocol indicated that HCQ should be prescribed to all (3%), certain (88%) or no patients (9%). Previous HCQ prescription, media pressure and the advice of colleagues also influenced HCQ prescription. Although 43.3% (95% CI, 38.5 to 48.1%) of the responders indicated that their HCQ prescription was influenced by the media pressure, that effect seemed somewhat variable: more prescriptions for 38.2% of the prescribers, fewer for 7.6% of them and 23.3% of non-prescribers. In contrast, no links were found between HCQ prescription and the intensity of the epidemic according to the region (S1 Fig, S4 Table), the physicians' sex, the duration of his/her practicing medicine or the type of hospital (university or not).

The multivariable analysis retained the following criteria as being significantly and independently associated with HCQ prescription (Table 3): an established departmental HCQ-prescription procedure (aOR, 8.25, 95% CI, 4.79 to 14.2; $P < 0.0001$), previously prescribed HCQ (aOR, 2.75, 95% CI, 1.5 to 5.03; $P = 0.001$), outside a therapeutic trial prescription of drugs other than HCQ to treat covid-19 (aOR, 3.21, 95% CI, 1.81 to 5.71; $P < 0.0001$).

## Discussion

This nationwide inquiry, conducted during the first wave of the covid-19 pandemic in France, addressing HCQ prescription, found that 45.3% of the responding hospitalists declared having prescribed HCQ to covid-19 patients outside any therapeutic trial. Two profiles could be discerned, distributed almost equally: prescribers, among whom only 13.8% based their decision on HCQ efficacy; and non-prescribers, among whom 85% relied on relevant evidence-based medicine (EBM) criteria to support their position. Our multivariate analysis retained the following as reasons being independently associated with prescribing HCQ: a departmental HCQ-prescription procedure, outside a therapeutic trial prescription of other treatments, prior HCQ prescription and no HCQ prescription within the framework of a therapeutic trial.

**Table 2. Physicians reasons for prescribing HCQ or not.**

| Reason | Hospitalists, n (%) | 95% CI† |
|---|---|---|
| **For prescription (multiple choice)** | **HCQ prescribers (N = 181)** | |
| Only therapeutic option available (no alternative) | 103 (56.9) | 49.6 to 63.9 |
| You applied the recommendation of the collegial decision | 92 (50.8) | 43.6 to 58 |
| It is an old drug with a known, favorable safety profile | 90 (49.7) | 42.5 to 56.9 |
| Covid-19 is a potentially very serious disease | 88 (48.6) | 41.4 to 55.9 |
| HCQ efficacy against covid-19 was not certain but usual rules for drugs are not applicable during a public health emergency | 64 (35.4) | 28.8 to 42.6 |
| Its prescription was made possible by a health ministry decree | 51 (28.2) | 22.1 to 35.1 |
| HCQ is an inexpensive and available drug | 51 (28.2) | 22.1 to 35.1 |
| Requested by the patient or his/her entourage | 37 (20.4) | 15.2 to 26.9 |
| HCQ seems to be effective against covid-19 | 25 (13.8) | 9.5 to 19.6 |
| It seemed difficult to resist media pressure | 17 (9.4) | 5.9 to 14.5 |
| It was recommended by colleagues | 13 (7.2) | 4.2 to 11.9 |
| It was recommended by "medical authorities" | 10 (5.5) | 3.0 to 9.9 |
| Fear of medical–legal consequences | 4 (2.2) | 0.9 to 5.5 |
| Fear of the how I would be viewed by my departmental colleagues | 4 (2.2) | 0.9 to 5.5 |
| The patient had already taken HCQ for another indication | 2 (1.1) | 0.3 to 3.9 |
| **Not to prescribe HCQ (multiple choice)** | **HCQ non-prescribers (N = 219)** | |
| No indication according to currently available medicine/science data | 163 (74.4) | 68.3 to 79.8 |
| Absence of official recommendation* supporting HCQ prescription | 111 (50.7) | 44.1 to 57.2 |
| Fear of potential adverse events | 79 (36.1) | 30.3 to 42.6 |
| You think it unethical to prescribe a non-validated drug outside therapeutic trials | 78 (35.6) | 29.6 to 42.2 |
| None of your patients met the criteria for HCQ prescription established in your department | 21 (9.6) | 6.4 to 14.2 |
| Fear that HCQ could contribute to covid-19 worsening | 12 (5.5) | 3.2 to 9.3 |
| You are opposed to off-label prescription | 10 (4.6) | 2.5 to 8.2 |
| No collegial organized discussion or in your department | 9 (4.1) | 2.2 to 7.6 |
| You didn't even consider its prescription | 7 (3.2) | 1.6 to 6.4 |
| Fear of medical–legal consequences | 5 (2.3) | 1.0 to 5.2 |

(*Continued*)

**Table 2.** (Continued)

| | | |
|---|---|---|
| Fear of the reactions or opinions of your colleagues | 4 (1.8) | 0.7 to 4.6 |
| Refusal of the patient or his/her entourage | 3 (1.4) | 0.5 to 3.9 |

* Learned societies, the Academy of Medicine, National Association of Physicians.

† IC 95% values are percentages of respondents.

## Percentage of HCQ prescribers

The observed prescriber rate (45.3%) was higher than in certain French (14%) [11] or international (12%) [19] inquiries, but lower than in others (90%) [12]. The differences can be explained by the specific prescription context (outpatient) [19], the country [12] or the time of the inquiries [19]. In the recurrent Sermo international inquiries, the mean prescriber rate was comparable to ours and varied over time: it increased from 33% to 58% from March to April 2020 (in-hospital or private practice), then declined from 66% to 28% from April to July (in-hospital practice exclusively) [10]. Those inquiries had methodological weaknesses: prescription context not always specified (in-hospital or private practice) [11], representativity of the responders unknown because of the selection method [10–12,19]. A high rate of HCQ prescription during the first wave of the pandemic has been associated by some authors with possible risks: toxicity [20–22], depletion of stocks [19,23] or deterring research [24,25]. Others suggest that HCQ may have been beneficial for patients hospitalized for Covid-19 (S1 Table). In any case, our study showed that French hospitalists were very divided on the prescription of HCQ during the first wave of covid-19 pandemic. This confirms the interest of an ethical reflection on the prescription of unproven interventions outside research in a pandemic period, such as that initiated by the WHO [26]. The examples against HCQ (unconfirmed efficacy as during Chikungunya virus infection) [3,27,28] and corticosteroids (efficacy discussed a priori [29], confirmed a posteriori) [30] illustrate the difficulty of this exercise.

## Physicians reasons for prescribing HCQ or not

The majority of prescribers seem to be aware that the effectiveness of HCQ has not been established: they used HCQ without citing efficacy as a criterion. They justifies their attitude by HCQ favorable tolerance profile or the disease severity, which are known prescription criteria, with others declared criteria: availability, price and regulatory context of HCQ prescription [31,32]. This attitude can evoke the prescription profile called "just do it" by Aquino and Cabrera in the context of the pandemic emergency, during which no specific treatment had proven efficacy [33]: prescribe treatments with unevaluated effects hoping for a favorable benefit/risk ratio but taking the risk of drugs being ineffective or even deleterious [34]. Furthermore, a clear majority (85%) of non-prescriber hospitalists explained their decisions citing relevant EBM criteria. A third of them declared they were afraid of potential adverse effects, seeming to adhere to the "first, do no harm" principle, even if that meant not prescribing a therapy that might later prove effective [33,34].

## Determinants of HCQ prescription

Our results confirmed that the HCQ-prescription rate differed according to the physician's specialty [35] or geographical region [10,15,19] but not according to the number of years of experience, unlike Baicus et al and Mehta et al [28,36]. But this association was not retained by multivariate analysis, not performed in others studies [10,15,19,35,36]. That finding held true

**Table 3. Univariate and multivariate analyses of HCQ-prescription determinants.**

| Determinant | Univariate analysis* | | Multivariate analysis* | |
|---|---|---|---|---|
| | aOR [95% CI] | P value | aOR [95% CI] | P value |
| Sex | | | | |
| Male | 1 | | | |
| Female | 0.75 [0.51 to 1.11] | 0.1526 | | |
| Years in practice, n | | | | |
| 0–4 | 1 | | | |
| 5–9 | 0.87 [0.45 to 1.7] | 0.6882 | | |
| 10–19 | 1.08 [0.57 to 2.03] | 0.8236 | | |
| 20–29 | 1.27 [0.68 to 2.38] | 0.4458 | | |
| $\geq$30 | 0.7 [00.35–1.39] | 0.3073 | | |
| Specialty | | | | |
| Other specialty | 1 | | | |
| Infectious diseases (ID) | 1.19 [0.69 to 2.04] | 0.5385 | | |
| Internal medicine (IM) | 2 [1.16 to 3.45] | **0.0121** | | |
| ID & IM | 2.72 [1.14 to 6.49] | **0.0245** | | |
| Hospital type | | | | |
| Private | 1 | | | |
| Teaching public | 0.61 [0.25 to 1.48] | 0.2771 | | |
| Non-teaching public | 0.76 [0.31 to 1.85] | 0.5419 | | |
| Hospital geographical region† | | | | |
| Paris region | 1 | | | |
| Northwest | 0.44 [0.23 to 0.83] | **0.0117** | | |
| Northeast | 0.43 [0.25 to 0.74] | **0.0021** | | |
| Southwest | 0.54 [0.26 to 1.1] | 0.0900 | | |
| Southeast | 1.07 [0.58 to 1.98] | 0.8348 | | |
| Previous HCQ prescription | | | | |
| No | 1 | | | |
| Yes | 2.14 [1.3 to 3.52] | **0.0029** | 2.75 [1.5 to 5.03] | 0.001 |
| HCQ procedure | | | | |
| No | 1 | | | |
| Yes | 8.36 [5.12 to 13.65] | **<0.001** | 8.25 [4.79 to 14.2] | < .0001 |
| Outside a therapeutic trial prescription (others/HCQ)§ | | | | |
| No | 1 | | | |
| Yes | 3.74 [2.3 to 6.07] | <0.001 | 3.21 [1.81 to 5.71] | < .0001 |
| HCQ prescription in therapeutic trial | | | | |
| No | 1 | | | |
| Yes | 0.68 [0.45 to 1.03] | 0.0707 | 0.56 [0.33 to 0.95] | 0.0301 |
| Other prescriptions in therapeutic trials | | | | |
| No | 1 | | | |
| Yes | 0.84 [0.56 to 1.28] | 0.4242 | | |
| Sensitive to media pressure | | | | |
| No | 1 | | | |
| Yes | 1.55 [1.04 to 2.32] | 0.0301 | | |

*Univariate analyses (P<0.2) selected potential explanatory variables that were then tested in the multivariate model (stepwise method with entry/stay significance levels of 0.2/0.05). The results are expressed as adjusted odd ratios (aOR) [95% confidence interval (CI)].

†Northwest: Normandy, Brittany, Centre-Val de Loire and Pays de la Loire; Northeast: Bourgogne-Franche-Comté, Grand Est and Hauts-de-France, Southwest: Nouvelle-Aquitaine and Occitanie; Southeast: Auvergne-Rhône-Alpes, Provence-Alpes-Côte d'Azur and Corsica.

§Lopinavir/ritonavir, Remdesivir, interleukin (IL)-6 inhibitors and/or IL-1 inhibitors, convalescent plasma, corticosteroids or others.

for the region and the specialty (internal medicine), probably partially because of the significant rate of internists who had previously prescribed HCQ before the covid-19 pandemic.

Among the determinants retained by the multivariate analysis, the existence of HCQ-prescription procedures was the factor the most strongly associated with HCQ prescription. Procedure existence per se could have incited hospitalists to prescribe HCQ. Moreover, departments that established procedures might have had more physicians favorable to HCQ use. The establishment of in-hospital procedures could enhance a sense of adequacy between healthcare practices and EBM [37,38]. No specific work on this question during a pandemic was found, other than a moderate-quality study of clinical guidelines published at the onset of the pandemic [39] and contribution of living guidelines [3].

The association between HCQ prescription in therapeutic trials and less HCQ prescription outside a therapeutic trial, as herein, could support the hypothesis that the HCQ prescription depends on the personality of the hospitalist: non prescribers are more likely to support EBM despite the pandemic context. But limiting that association to therapeutic trials on HCQ might simply illustrate that, for the departments that established therapeutic trials on HCQ, the hospitalists preferred including patients in trials rather than prescribe outside one. Moreover, the fact that HCQ prescribers declared prescribing, more often as non-prescribers, other treatments of non-proven efficacy against covid-19 indicates that their prescribing attitude is not specific to HCQ and to a more general tendency to prescribe treatments with unproven efficacy in times of pandemic, adopting a "just do it" profile [33,34]. Doctors experienced in using HCQ for another indication had more HCQ prescriptions for covid-19. Indeed, prior experience with a drug is a known factor for prescription [31]. This finding could reflect a cognitive bias [40].

While the link between poor-quality medical publications, media repercussions, and HCQ-prescription policies [8,13,15,35,41,42] or more HCQ prescriptions [13,15,35] has been reported, our multivariate analysis did not retain media pressure as being associated with increased HCQ prescription.

## Strengths and limitations of this study

To our knowledge, no other study has evaluated the HCQ-prescription rate and reasons for its prescription during the pandemic of a well-characterized population of French internal medicine or infectious disease hospitalists. Moreover, responders estimated the mean overall quality of the questionnaire to be 7.2 (±1.2)/10.

The timing of the inquiry (May 2020) during the first wave of the covid-19 pandemic in France (February to May 2020) enabled assessment of HCQ prescription. This choice might also have biased certain responses. However, after the inclusion period no randomized trial results on a large population were available [43] and both French and World Health Organization (WHO) public health authority released recommendations toward HCQ use to treat covid-19 only within therapeutic trial [44,45].

Our study has several limitations. The response rate was only 22.8%, but nevertheless close to other inquiries on HCQ use during the pandemic: 17% (1215/7000) [12], 29% (785/2645) [36], or 27% (71/260) [46]. Our study was declarative, which could have partially impacted the results. Our inquiry was sent exclusively to internal medicine and infectious disease departments, so the findings cannot be extrapolated to other departments that could have prescribed HCQ, eg, geriatrics or intensive care.

## Conclusions

This study provides information on in-hospital HCQ prescription outside any therapeutic trial in France during the first wave of the covid-19 pandemic: its high frequency, varied practices

of the hospitalists according to each one's questioning of EBM principles because of the pandemic, the influence of codified hospital procedures. Understanding the modalities of hospitalists' therapeutic decision-making during the pandemic would be a first step towards subsequently optimizing therapeutic decision-making processes. The existence of prescription procedures was the factor the most strongly associated with HCQ prescription herein. Nonetheless, our findings suggest that prescription procedures during the pandemic is a way to improve the quality of therapeutic decisions. Notably, such protocols, elaborated by independent authorities, based on robust scientific data and up-dated according to validated procedures could help physicians provide better care of their patients.

## Supporting information

**S1 Fig. Correlation between the in-hospital hydroxychloroquine prescription rate and cumulative in-hospital–mortality rate according to French geographic region during the first semester of 2020, ie, the first wave of covid-19.**
(DOCX)

**S1 Table. Observational studies of off-label HCQ prescription for hospitalized covid-19 patients in real-life.**
(DOCX)

**S2 Table. Comparison of participants' specialties.**
(DOCX)

**S3 Table. HCQ-prescription determinants: Sources of information on HCQ to treat covid-19.**
(DOCX)

**S4 Table. Cumulative in-hospital–mortality rate per 100,000 inhabitants linked to covid-19 on May 7, 2020 in France.**
(DOCX)

**S1 Appendix. Inquiry questionnaire (French).**
(DOCX)

**S2 Appendix. Inquiry questionnaire (English).**
(DOCX)

**S3 Appendix. Checklist for Reporting Results of Internet E-Surveys (CHERRIES).**
(DOCX)

**S1 File.**
(XLSX)

## Acknowledgments

We sincerely thank all the doctors who took the time to complete the questionnaire. The authors also thank Pierre Tattevin, president of the French Infectious Diseases Society (SPLIF), for having accepted to send the questionnaire via email to Society members. The authors thank Stéphanie Rouanet (StatEthic SASU) for the statistical analyses and Janet Jacobson for editorial assistance.

## Author Contributions

**Conceptualization:** Antoine Bosquet, Comlan Affo, Isabelle Mahé.

**Data curation:** Antoine Bosquet.

**Formal analysis:** Antoine Bosquet, Comlan Affo, Isabelle Mahé.

**Investigation:** Antoine Bosquet.

**Methodology:** Antoine Bosquet, Comlan Affo, Ludovic Plaisance, Isabelle Mahé.

**Project administration:** Antoine Bosquet.

**Resources:** Antoine Bosquet.

**Supervision:** Antoine Bosquet.

**Validation:** Antoine Bosquet, Comlan Affo, Emmanuel Mortier, Isabelle Mahé.

**Visualization:** Antoine Bosquet, Isabelle Mahé.

**Writing – original draft:** Antoine Bosquet, Comlan Affo, Emmanuel Mortier.

**Writing – review & editing:** Antoine Bosquet, Comlan Affo, Ludovic Plaisance, Géraldine Poenou, Emmanuel Mortier, Isabelle Mahé.

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
