## [Decision Letter · Decision Letter 0]

2 Nov 2021

PONE-D-21-20849Off-label prescription of hydroxychloroquine for hospitalized patients with covid-19: a national inquiry of prescription patterns among French hospitalistsPLOS ONE

Dear Dr. Antoine Bosquet,

Thank you for submitting your manuscript to PLOS ONE. After careful consideration, we feel that it has merit but does not fully meet PLOS ONE’s publication criteria as it currently stands. Therefore, we invite you to submit a revised version of the manuscript that addresses the points raised during the review process.

We look forward to receiving your revised manuscript.

Kind regards,

Abdelwahab Omri, Pharm B, Ph.D

Academic Editor

PLOS ONE

“Dr. Mahé reports grants from BMS Pfizer, grants from LEO Pharma, personal fees and non-financial support from BMS Pfizer, personal fees and non-financial support from Leo Pharma, personal fees and non-financial support from Bayer, outside the submitted work.”

Reviewers' comments:

Reviewer's Responses to Questions

**Comments to the Author**

1. Is the manuscript technically sound, and do the data support the conclusions?

Reviewer #1: Yes

Reviewer #2: Yes

2. Has the statistical analysis been performed appropriately and rigorously? 

Reviewer #1: N/A

Reviewer #2: Yes

3. Have the authors made all data underlying the findings in their manuscript fully available?

Reviewer #1: No

Reviewer #2: Yes

4. Is the manuscript presented in an intelligible fashion and written in standard English?

Reviewer #1: Yes

Reviewer #2: Yes

5. Review Comments to the Author

Reviewer #1: This manuscript is interesting because it reports original data concerning the prescription of hydroxychloroquine. So from the last method of results is interesting.

The discussion is too biased from my point of view, firstly there are many studies showing that off-label prescribing in French hospitals is up to 70% and therefore it is likely that the physicians who responded did not take this into account or do not really know what off-label drugs are, globally those off-label, that is, prescriptions not recognized by the "Agence Française du Médicament" do not necessarily correspond to what is recognized by the FDA, or those considered ineffective by the Cochrane Library The notion of off-label is at least as political a notion as it is scientific and country dependent.

The authors conclude that the unanimity made sure on the toxicity, and on the ineffectiveness of hydroxychloroquine, which is partisan and untrue. It does not really add value to this article and does not reflect the nature of the data in the literature

This part is therefore a partisan part for me and is too long and not linked to the study. The discussion should be limited to an evaluation of the answers and a reflection on the direction of answers, given the wording of the questions that are asked, this is a phenomenon usually observed in all surveys. Moreover, the notion of media pressure is a subjective notion and I suggest that this data be eliminated. In practice, this work is important, but the discussion reflects a partisan vision that profoundly reduces the value of the work, which should be concentrated on the observational part that is related to it.

Reviewer #2: This article is written well. However, I observed some discrepancies in the article. I would like to encourage authors to address the following points

1. Rewrite the abstract part in an organized manner

2. Study objective is not a part of the Method. Included this section in the introduction section

3. I found some general sentences and heading in the method section. That needs to be removed from there. They can be included in the introduction section (Declaration to the French 108 Computer Watchdog Commission (CNIL), Patient and public involvement)

4. Table 2 and Table 3 showed the 95% CI value. However, there are discrepancies between the range with their mean. Please reverify the calculation.

6. PLOS authors have the option to publish the peer review history of their article (what does this mean?). If published, this will include your full peer review and any attached files.

Reviewer #1: No

Reviewer #2: **Yes: **Sujit Kumar Debnath

---

## [Author Response · Author response to Decision Letter 0]

9 Dec 2021

Colombes, France, December 9th, 2021

Dear Editor,

 Please find on the PLOS ONE submission web site (https://www.editorialmanager.com/pone/) the revised version of our manuscript ID entitled, Off-label prescription of hydroxychloroquine for hospitalized patients with covid-19: a national inquiry of prescription patterns among French hospitalists, by A Bosquet, C Affo, L Plaisance, G Poenou, E Mortier and I Mahé. We took into account the requests of the PLOS ONE Academic Editor and reviewers 1 and 2.

 Please find below the answers to each point raised by academic editor and reviewers. 

1/Academic requirements

- Academic editor requirement 1. “Please ensure that your manuscript meets PLOS ONE's style requirements, including those for file naming.”

I made sure our manuscript meets PLOS ONE’s style requirements and modify for example heading and subheading styles or add a table (S4 table: line 608) to prevent a table from being in the legend of figure 1 in supporting information section.

- Academic editor requirements 2. “(…) Thank you for stating the following in the Competing Interests section:“Dr. Mahé reports grants from BMS Pfizer, grants from LEO Pharma, personal fees and non-financial support from BMS Pfizer, personal fees and non-financial support from Leo Pharma, personal fees and non-financial support from Bayer, outside the submitted work. Please confirm that this does not alter your adherence to all PLOS ONE policies on sharing data and materials, by including the following statement: ""This does not alter our adherence to PLOS ONE policies on sharing data and materials.” (as detailed online in our guide for authors http://journals.plos.org/plosone/s/competing-interests). If there are restrictions on sharing of data and/or materials, please state these. Please note that we cannot proceed with consideration of your article until this information has been declared.

Please include your updated Competing Interests statement in your cover letter; we will change the online submission form on your behalf.”

Dr Mahé and I confirm that this does not alter our adherence to all PLOS ONE policies on sharing data and materials: "This does not alter our adherence to PLOS ONE policies on sharing data and materials.”

2/ General comments from reviewers

- Comments 1 “(Is the manuscript technically sound, and do the data support the conclusions? Yes/yes”) and 2 (“Has the statistical analysis been performed appropriately and rigorously? (yes/yes”). 

N/A

- Reviewers’ comment 3 (“Have the authors made all data underlying the findings in their manuscript fully available? Reviewer #1 answer: No”)

We agree to communicate the raw data of our work (answers from the 400 hospitalists who answered the questionnaire). The file name is: CovHYd Study (answers) Plos ONE_20211126_n400.

- Reviewers’ comment 4 (“Is the manuscript presented in an intelligible fashion and written in standard English? Yes/yes”). 

N/A

3/ Additional comments from reviewer#1 

“This manuscript is interesting because it reports original data concerning the prescription of hydroxychloroquine. So from the last method of results is interesting” 

We thank the reviewers for their careful reading and their relevant and helpful comments.

“The discussion is too biased from my point of view, firstly there are many studies showing that off-label prescribing in French hospitals is up to 70% and therefore it is likely that the physicians who responded did not take this into account or do not really know what off-label drugs are, globally those off-label, that is, prescriptions not recognized by the "Agence Française du Médicament" do not necessarily correspond to what is recognized by the FDA, or those considered ineffective by the Cochrane Library The notion of off-label is at least as political a notion as it is scientific and country dependent.”

Thank you for this remark. Indeed, the term "off label prescription" used several times in the article is not the correct one. This is a translation error. We suggest replacing it generally with "outside a (or any) therapeutic trial" or in line 265 in marked copy manuscript with "unproven interventions outside of research", the term used in the WHO report (new reference 26). 

“The authors conclude that the unanimity made sure on the toxicity, and on the ineffectiveness of hydroxychloroquine, which is partisan and untrue. It does not really add value to this article and does not reflect the nature of the data in the literature. This part is therefore a partisan part for me and is too long and not linked to the study. The discussion should be limited to an evaluation of the answers and a reflection on the direction of answers, given the wording of the questions that are asked, this is a phenomenon usually observed in all surveys.”

Indeed, the subject of the study is whether hospitalists prescribed HCQ and if yes or not, for what reasons during the first wave of the Covid-19 pandemic (when the benefit / risk ratio of HCQ was uncertain in this case). The aim of the study is not to know whether HCQ is effective, or not, in treating covid-19. We therefore propose to refocus the discussion on the answers given by the doctors and their significance. In addition, we have modified some formulations to be more neutral: for example, 

- line 260 (marked copy manuscript) we wrote a high rate of HCQ prescription (…) has been associated by some authors with possible risk (insteed of … is associated with possible risk …), 

- line 262, we have added “Others suggest that HCQ may have been beneficial for patients hospitalized for Covid-19 (S1 table)”, 

- line 323, we have removed the sentence “Almost half of the hospitalists responding declared having prescribed HCQ without the efficacy of this molecule being recognized for the treatment of covid-19”

- line 353, we have removed the following words “ and is not without risk (29-31)”

- lines 368-370, we have removed the sentence “.In May 2020, studies, often retrospective and observational, going against HCQ use accumulated”

“Moreover, the notion of media pressure is a subjective notion and I suggest that this data be eliminated.” 

Indeed, we completely agree that media pressure is a very subjective notion, that it is complex to analyze the responses that refer to it. We therefore propose to communicate the raw results concerning media pressure and to remove interpretations that could be risky. For example, we have removed the long discussion on media pressure, the HCQ prescription authorization policies and the prescription of HCQ (lines 381-402), the words “ways healthcare professionals can resist media pressure” (line 424), or “media pressure” (line 435 and 448) in the discussion section. In abstract, we propose to remove the sentence “Media pressure influenced HCQ prescription by 43.3% of the responders” (line 44-45) and the word “media pressure” (line 52).

“In practice, this work is important, but the discussion reflects a partisan vision that profoundly reduces the value of the work, which should be concentrated on the observational part that is related to it.”

As we indicated above, we have corrected the discussion to be closer to the answers of the hospitalists and revised some formulations to be more objective.

4/ Additional comments from reviewer #2 

Reviewer #2: This article is written well. However, I observed some discrepancies in the article. I would like to encourage authors to address the following points

1. Rewrite the abstract part in an organized manner 

We still completely agree with this remark and have followed the indications given on the following sites: https://journals.plos.org/plosone/s/submission-guidelines and https://plos.org/resource/how-to-write-a-great-abstract/.

2. Study objective is not a part of the Method. Included this section in the introduction section 

We removed this section in the methods section and included it at the end of the introduction.

3. I found some general sentences and heading in the method section. That needs to be removed from there. They can be included in the introduction section (Declaration to the French 108 Computer Watchdog Commission (CNIL), Patient and public involvement)

We proposed to remove the sentence about patient and public involvement as they were not involved due to the pandemic context and the study protocol. On the other hand, we propose to keep the sentence concerning the declaration of the study to the French commission of the CNIL because it is a regulatory point of protection of people who answered the questionnaire of our study. This declaration is compulsory in France for this kind of study. It does not seem to us to have its place in the introduction according to the indications of the PLOS ONE submission guidelines site (introduction section: https://journals.plos.org/plosone/s/submission-guidelines).

4. Table 2 and Table 3 showed the 95% CI value. However, there are discrepancies between the range with their mean. Please reverify the calculation.

In table 2 (page 9), 95% IC values are percentages of respondents, not absolute numbers of respondents. We therefore propose to specify in a caption this "IC 95% values are percentages of respondents" in order to avoid any confusion (line 173). In Table 3, the values indicated are only odds ratios and their 95% IC. We do not believe there is any confounding factor. We checked the data of Table 3 for accuracy and found no errors. We therefore propose not to modify the original text.

To finish, changes to the discussion resulted in changes to the bibliography (46 references remaining instead of the initial 63). We have also made some additional changes:

- we find a mistake in bibliography (wrong reference 44 in Marked copy manuscript, line 589) and propose to correct it: “HCSP. SARS-CoV-2: therapeutic recommendations. Rapp HCSP. Paris: Haut Conseil de la Santé Publique; 2020 MarJun. Available: https://www.hcsp.fr/Explore.cgi/avisrapportsdomaine?clefr=801
https://www.hcsp.fr/Explore.cgi/AvisRapportsDomaine?clefr=847 »

- we have moved the discussion sub-chapter "strengths and weaknesses of this study" at the end of the discussion as is most often the custom.

 We hope you will consider our paper for publication in Plos ONE, and we look forward to hearing from you.

Sincerely,

Antoine Bosquet, MD, for the authors

---

## [Editor Report · Decision Letter 1]

13 Dec 2021

Outside any therapeutic trial prescription of hydroxychloroquine for hospitalized patients with covid-19 during the first wave of the pandemic: a national inquiry of prescription patterns among French hospitalists

PONE-D-21-20849R1

Dear Dr. Antoine Bosquet,

We’re pleased to inform you that your manuscript has been judged scientifically suitable for publication and will be formally accepted for publication once it meets all outstanding technical requirements.

Kind regards,

Abdelwahab Omri, Pharm B, Ph.D, 

Academic Editor

PLOS ONE

---

## [Editor Report · Acceptance letter]

22 Dec 2021

PONE-D-21-20849R1 

Outside any therapeutic trial prescription of hydroxychloroquine for hospitalized patients with covid-19 during the first wave of the pandemic: a national inquiry of prescription patterns among French hospitalists 

Dear Dr. Bosquet:

I'm pleased to inform you that your manuscript has been deemed suitable for publication in PLOS ONE. Congratulations! Your manuscript is now with our production department. 

Kind regards, 

on behalf of

Dr. Abdelwahab Omri 

Academic Editor

PLOS ONE